# The Association between Serum Levels of 25[OH]D, Body Weight Changes and Body Composition Indices in Patients with Heart Failure

**DOI:** 10.3390/jcm9041228

**Published:** 2020-04-24

**Authors:** Apolonia Stefaniak, Robert Partyka, Sylwia Duda, Weronika Ostręga, Jacek Niedziela, Jolanta Nowak, Jolanta Malinowska-Borowska, Tomasz Rywik, Przemysław Leszek, Bartosz Hudzik, Barbara Zubelewicz-Szkodzińska, Piotr Rozentryt

**Affiliations:** 1Department of Toxicology and Health Protection, Faculty of Health Sciences in Bytom, Medical University of Silesia in Katowice, 41-902 Bytom, Poland; sduda@sum.edu.pl (S.D.); weronika.ostrega@gmail.com (W.O.); jmalinowska@sum.edu.pl (J.M.-B.); prozentryt@sum.edu.pl (P.R.); 2Clinical Division of Anesthesiology and Intensive Therapy of the Department of Anesthesiology, Intensive Treatment and Emergency Medicine, Medical University of Silesia, 41-800 Zabrze, Poland; robertpartyka@op.pl; 3Department of Cardiology, Faculty of Medical Sciences in Zabrze, Medical University of Silesia, Silesian Centre for Heart Disease, 41-800 Zabrze, Poland; jacek.niedziela@gmail.com (J.N.); nowjola@wp.pl (J.N.); bartekh@mp.pl (B.H.); 4Heart Failure and Transplantology Department The Cardinal Stefan Wyszynski Institute of Cardiology, 04-628 Warsaw, Poland; t.rywik@ikard.pl (T.R.); przemyslaw.leszek@ikard.pl (P.L.); 5Department of Cardiovascular Disease Prevention, Faculty of Health Sciences in Bytom, Medical University of Silesia in Katowice, 41-902 Bytom, Poland; 6Department of Nutrition-Related Disease Prevention, Department of Metabolic Disease Prevention, Faculty of Health Sciences in Bytom, Medical University of Silesia in Katowice, 41-902 Bytom, Poland; basiazub@poczta.onet.pl

**Keywords:** heart failure, body wasting, weight change, 25[OH]D, metabolic instability

## Abstract

We try to determine the association between weight changes (WC), both loss or gain, body composition indices (BCI) and serum levels of 25[OH]D during heart failure (HF). WC was determined in 412 patients (14.3% female, aged: 53.6 ± 10.0 years, NYHA class: 2.5 ± 0.8). Body fat, fat percentage and fat-free mass determined by dual energy X-rays absorptiometry (DEXA) and serum levels of 25[OH]D were analyzed. Logistic regression was used to calculate odds ratios for 25[OH]D insufficiency (<30 ng/mL) or deficiency (<20 ng/mL) by quintiles of WC, in comparison to weight-stable subgroup. The serum 25[OH]D was lower in weight loosing than weight stable subgroup. In fully adjusted models the risk of either insufficient or deficient 25[OH]D levels was independent of BCI and HF severity markers. The risk was elevated in higher weight loss subgroups but also in weight gain subgroup. In full adjustment, the odds for 25[OH]D deficiency in the top weight loss and weight gain subgroups were 3.30; 95%CI: 1.37–7.93, *p* = 0.008 and 2.41; 95%CI: 0.91–6.38, *p* = 0.08, respectively. The risk of 25[OH]D deficiency/insufficiency was also independently associated with potential UVB exposure, but not with nutritional status and BCI. Metabolic instability in HF was reflected by edema-free WC, but not nutritional status. BCI is independently associated with deficiency/insufficiency of serum 25[OH]D.

## 1. Introduction

Heart failure (HF) is a clinical syndrome characterized by numerous neuroendocrine alterations and inflammation both leading to elevated resting metabolic rate, catabolism, progressive multiple organ dysfunction and poor prognosis [1]. Despite modern treatment, tissue loss from all body compartments is recognized in most of HF patients [2], while in some cases weight gain is also observed [3]. Current understanding of wasting in HF is based on the concept of catabolic/anabolic imbalance. Predominance of catabolic signals in HF is generally accepted as a main mechanism of muscle and fat wasting [4]. 

It has been shown in the studies that vitamin D is implicated into regulation of mitochondria, energy expenditure and oxygen consumption [5,6,7]. Additionally, the number of physiological pathways important for structural and functional integrity of muscle and adipose tissue are under the influence of vitamin D signaling. Studies in animals have confirmed that the presence of the functional vitamin D receptor (VDR) and some enzymes are necessary to metabolize the vitamin D to its active hormonal form, 1,25(OH)_2_D, both in myocytes and in adipocytes [8,9]. VDR stimulation is important for expression of genes associated with myotubules synthesis [10,11] but also for metabolism of adipose tissue [12,13]. For example, VDR knock-out mice have lower body weight; their muscle fibers are 20% smaller as compared to wild-type animals. Additionally, they show less expression of several myogenic factor and myocytes are highly variable in size [14]. Mouse skeletal muscle myocytes regenerating after injury show higher expression of VDR in comparison to myocytes with no signs of regeneration [15]. In mice, VDR removal by genetic engineering leads to fat loss despite higher compensatory food intake [12]. Conversely, adipose specific VDR overexpression in tissue reduces basal metabolism and increases fat mass [13]. 

Additionally, several observations and experiments in humans confirm reports received on animals. In aging humans, VDR in muscle cell decreases [16], while vitamin D supplementation in a randomized trial in women with insufficient levels of vitamin D demonstrated a 30% increase in expression of VDR and 10% increase of muscle mass [17]. These cellular, animal and human studies demonstrate the importance of VDR mediated signaling for proper structure and function of myocytes and adipocytes. In HF, the association with low vitamin D levels have been repeatedly reported [18,19]. However, the relationship between vitamin D levels and muscle and fat content in the context of HF-induced WC has not been studied yet.

Therefore, the aim of the study was to analyze the association between degree of anabolism/catabolism represented by WC, muscle and fat mass and serum levels of 25[OH]D in stable HF patients. 

## 2. Materials and Methods

### 2.1. Study Group

Data collected in Prospective Registry of Heart Failure (PRHF) implemented in our Department since the end of 2003 were used in the study. The Registry was dedicated to patients who were considered potential candidates for heart transplantation and from its onset comprised more than 1200 patients. The main inclusion criterion for the current analysis was the availability of serum 25[OH]D level measurements performed from samples collected at a time of inclusion into the Registry. We selected patients with HF and reduced left ventricle ejection fraction (LVEF ≤ 40%), diagnosed according to European Society of Cardiology criteria, above 18 years of age. PRHF patients were recruited in Outpatient Clinic between January 2004 and March 2013. We included patients who reached maximum tolerated doses of recommended drugs, in whom the onset of HF could be identified with one-month precision and whose records of body weight before first diagnosis of HF were available. The onset of HF was defined as a month in which medical records prepared by cardiologist in ambulatory settings showed coexistence of LVEF ≤ 40% with typical signs and/or symptoms of HF. 

The pre-HF body weight was defined based on medical records of Outpatients Clinic as the highest weight within a year, but not later than 2 months before diagnosed HF. On the day of inclusion into the study the weight was measured and defined as index weight (dry, edema-free weight) only if attending cardiologist during clinical examination neither changed diuretic therapy nor recorded symptoms of fluid retention.

We excluded patients treated with glucocorticosteroids, bisphosphonates, vitamin D, calcium or phosphorus salts as separate drugs or as multivitamins and those having active infection, liver disease with enzymes four times higher than normal, active bleeding, known neoplasm or granulomatous disease, patients with a history of reduced absorptive capacity due to bariatric surgery. In total, 412 Registry participants met the study criteria and their data were included in the final analysis. 

Comorbidities, such as hypertension, diabetes mellitus and hypercholesterolemia, were recognized based on clinical history, current medication or actual measurements of respective variables. History of smoking was defined as current or previous use of tobacco products.

Blood samples were drawn in a standardized manner in the morning, between 8 and 10 AM, from patients who had been fasting for at least 8 h and resting in supine position in a quiet, environmentally controlled room for 30 min. Blood was immediately centrifuged in 4 °C and stored at −75 °C for further analyses. All procedures were undertaken in accordance to Helsinki Declaration. The protocol was reviewed and accepted by Ethical Committee of Medical University of Silesia. All patients have expressed their informed, written consent.

### 2.2. Measurements 

Body mass and height were measured on a day of blood sampling (index date) using a certified scale (B150L, Redwag, Zawiercie, Poland). 

Body mass index (BMI) was calculated by dividing weights in kilograms by height in meters squared. BMI was calculated based on body weight on the index day (index BMI) and before heart failure (pre-HF). WC was calculated from the Formula (1) shown below:(1)$$WC\;\left(\%\right)=\frac{preHF\;BMI-index\;BMI}{preHF\;BMI}\cdot 100\%$$

A Sonos-5000 Hewlett-Packard Ultrasound Scanner; (Hewlett-Packard, Andover, MA, USA) was used to measure LVEF from the apical four-chamber view. LVEF was calculated from the Formula (2): (2)$$LVEF\;\left(\%\right)=\frac{\mathrm{end}-\mathrm{diastolic}\;\mathrm{volume}\;-\;\mathrm{end}-\mathrm{systolic}\;\mathrm{volume}}{\mathrm{end}-\mathrm{diastolic}\;\mathrm{volume}}\cdot 100\%$$

Body composition analysis was performed with the use of dual X-ray absorptiometry (DEXA) with a pencil beam Lunar DRX-L device (General Electric, Brussels, Belgium). We measured the total fat mass, percent of fat and fat-free mass used as a proxy for muscle mass in further analyses. Commercially available reagents (Roche Diagnostics, Switzerland) allowed to measure serum creatinine, C-reactive protein, cholesterol, number of albumins, lymphocytes and calcium. Global nutritional status was evaluated with the use of integrated score known as CONUT (CONtrolling NUTritional status), based on serum albumin, lymphocytes counts and serum cholesterol [20]. Glomerular filtration rate (GFR) was calculated from MDRD Formula (3): (3)$$eGFR\left(MDRD\right)=186.3\cdot plasma\;creatinine^{-1.154}\cdot age^{-0,203}\cdot 0.742\left(if\;female\right)$$

In the case of serum albumin <40 g/L calcium was corrected according to the fourth Formula (4):(4)$$corrected\;calcium\;\left[\frac{mmol}{L}\right]=total\;calcium+0.02\cdot \left(40-plasma\;albumin\;\left[\frac{g}{L}\right]\right)$$

For all but 25[OH]D and N-terminal pro-brain natriuretic peptide (NTproBNP), laboratory procedures as well as DEXA scanning and LVEF measurement were performed on the day of signing consent to participate in the PRHF (index date). The analyses of serum 25[OH]D and NTproBNP were undertaken in September 2011 taking advantage of commercially available diagnostic kits (Roche Diagnostics, Switzerland). In case of 25[OH]D, we used the Elecsys competitive immunoassay for the detection of total 25[OH]D (catalogue number 05894913190). According to characteristics provided by manufacturer Elecsys 25[OH]D, the total assay allows measurement of serum levels between 3–75 ng/mL. Liquid Chromatography Tandem-Mass Spectrometry (LC-MS) is considered as the gold standard technique with the capability for reliable and accurate quantification of circulating vitamin D metabolites. However, direct measurement of Serum Total Vitamin D (25-OH) Using Immunoassay in Comparison to LC-MS revealed very good agreement between them [21]. [21]. Serum levels of NTproBNP were measured using monoclonal electrochemiluminescence immunoassays (catalogue number 04842464190), which provided valid measurement of NTproBNP in a range of 5−35,000 pg/mL. Before analyses used in the current study, samples were deep-frozen for median time of 63 months, interquartile range 18 months and minimal and maximal time of 23 and 84 months, respectively.

Based on the literature serum levels, 25[OH]D < 30 ng/mL were defined as insufficient, whereas levels below 20 ng/mL as deficient [22]. The potential UVB exposure (UVBE) was approximated according to the date of blood sampling for 25[OH]D analysis. If blood sampling took place from October to the end of March, UVB exposure was considered low, while from April to September high, respectively. This division was based on the demonstration that too low UVBE between October to March in a population living in Europe at an altitude similar to Poland was responsible for ineffective skin production of vitamin D_3_ [23].

Study flow is presented on Figure 1. 

### 2.3. Statistical Analysis

Continuous variables are presented as mean values and standard deviations, categorical as percentages. Non-normally distributed data (Shapiro-Wilk testing) are presented as median and interquartile ranges (IQR). Analyses with non-normally distributed data were performed with log (base 10) transformed values (in tables shown before transformation). Spearman correlation was used to explore potential associations between variables. We constructed a quadratic polynomial approximation to check for a potential nonlinear association between WC and 25[OH]D deficiency/insufficiency. The study group was split into quintiles of WC and compared using Kruskal-Wallis or chi-square tests where appropriate. Furthermore, a multiply comparison analysis between subgroups of WC was done. To obtain standardized BMI and body composition indices, we subtracted the mean value from each individual value of a given parameter and then divided it by its standard deviation. After such a transformation, we received standardized BMIs, fat and fat-free tissue contents in which the mean values were zero with a standard deviation of one. The transformation allowed graphical demonstration of dynamic changes of BMI, fat and fat-free compartments at each quintile of WC. We took advantage of logistic regression to estimate the risk of 25[OH]D insufficiency/deficiency at each quintiles of WC. We have selected quintile 2 with the smallest WC during HF as a reference and calculated unadjusted odds of 25[OH]D deficiency/insufficiency at consecutive quintiles of WC. In the next step the effect of potential confounders including body composition indices and nutritional biomarkers was taken into account in estimation of odds ratio. For all analyses, significance level was set at 0.05 (two-tailed) and all calculations were performed using software packages of “Statistica” v.10.0 and NCSS v2007.

## 3. Results

### 3.1. The Basic Characteristics of Study Group and Comparisons between WC Subgroups

Patients in WC subgroups did not differ according to sex, age, etiology and duration of HF, potential exposure to UVB, MVO_2_ expressed per kg of fat-free mass, serum cholesterol and kidney function (Table 1). 

In higher quintiles of WC, patients had higher BMI before HF onset (*p* < 0.0001), they lost more weight (*p* < 0.0001), and had lower index BMI (*p* < 0.0001). They also showed more advanced NYHA class (*p* < 0.0001), lower left ventricle ejection fraction (*p* < 0.0001), worse exercise capacity as measured by MVO_2_, (*p* = 0.01), higher serum phosphate (*p* = 0.001), calcium (*p* < 0.0001), hsCRP (*p* = 0.002) and NTproBNP (*p* < 0.0001). Finally, in accordance to their worse risk profile, 1-year mortality in these patients peaked in top quintile of WC (*p* = 0.04) (Table 1).

The patients in higher quintile of WC—who lost more weight—had lower levels of 25(OH)D (*p* = 0.01) and serum levels of this compound were more frequently classified as deficient (25[OH]D <20 ng/mL) (*p* = 0.004), but not as insufficient (25[OH]D < 30 ng/mL) (*p* = 0.11) (Table 1). 

Nutritional status as reflected by the number of blood lymphocytes was worse in higher quintiles of WC (*p* = 0.047), but was not different if shown by serum albumin level (*p* = 0.11). The integrated CONUT index revealed only a trend for worse nutrition in patient at higher quintiles of WC (*p* = 0.07) (Table 1). 

The profile of comorbidities was similar in subgroups of WC. The administration rate of key HF drugs was comparable; only aldosterone antagonist (*p* = 0.004), loop diuretics (*p* = 0.004) and digoxin (*p* < 0.0001) were given more often to patients losing more weight during HF (Table 1). 

The percentage of fat was the highest among patients with weight gain (Q1) (*p*< 0.0001), and steadily dropped down in higher quintiles of WC in parallel to BMI decline (Table 1, Figure 2). In contrast to changes of BMI and fat mass, fat-free tissue was relatively preserved and a more visible decrease is only seen in the top quintile of WC (*p* = 0.0002) (Table 1, Figure 2). The standardized values of BMI, fat and fat-free mass according to quintiles of WC are shown in Figure 2.

Serum levels of 25[OH]D correlated with WC in pooled subgroups with weight loss (Q3 to Q5) (*r* = 0.16, *p* < 0.05), but not in patients with weight gain (*r* = 0.09, NS). There was a nonlinear association between WC and 25[OH]D deficiency/insufficiency, with the lowest probability of these abnormalities occurring in patients with no WC and an increase toward both extremes of WC. Figure 3 shows polynomial approximation of 25[OH]D deficiency probability across whole spectrum of WC.

### 3.2. The Risk of 25(OH)D Insufficiency/Deficiency in Relation to Weight Change

In unadjusted model the risk of 25[OH]D deficiency was more than doubled in patients who lost more than approximately 10% of their weight in comparison to patients with stable weight (Q4 and Q5 of WC, Table 2), and this risk increase was statistically significant. The odds in patients with weight gain was not different from the reference subgroup. The same analysis performed for 25[OH]D insufficiency showed less pronounced risk in weight-losing subgroups, significant only in quintile Q5. The risk in patients gaining weight was similar to the reference subgroup (Table 2).

After model correction, taking into account differences in potential UVBE, CONUT index, fat mass and fat-free mass, the odds for 25[OH]D deficiency increased and remained highly significant in subgroups with weight loss (Q4 and Q5), but were also elevated on the border of significance in subgroup with weight gain (Q1) (odds = 2.47; 95% CI: 0.94–6.53, *p* = 0.06, Table 2). When we replaced fat mass with fat percentage, the model did not change significantly.

In case of 25[OH]D insufficiency the risk associated with weight loss was less visible and only significant in the top weight loss (Q5), but the risk in weight gain subgroup increased slightly and become significant with odds = 2.53; 95% CI: 1.07–6.01, *p* = 0.03 (Table 2).

In the next step, we constructed fully adjusted models in which odds for 25[OH]D deficiency or insufficiency at quintiles of WC were corrected also with age, NYHA class (or NTproBNP or MVO_2,_ either expressed per kg of body mass or per kg of fat-free tissue), potential UVBE, CONUT, fat mass (or fat percentage) and fat-free mass. 

For both 25[OH]D deficiency or insufficiency, correction with age and markers of HF severity did not alter significantly the risk estimations. Again, the highest risk was shown in weight losing subgroups, significant in Q4 and Q5 for 25[OH]D deficiency and only in Q5 for 25[OH]D insufficiency. In patients gaining weight, for both 25[OH]D deficient and insufficient groups the risk was consistently elevated with only a trend toward significance (*p* = 0.08 and *p* = 0.07) (Table 2). Neither replacement of NYHA for NTproBNP or MVO_2_, nor fat mass to fat percentage significantly changed the risk values 

In the final models, the only significant predictors of 25[OH]D deficiency/insufficiency was potential UVBE and higher quintiles of WC. In patients with weight gaining, the risk was elevated with a trend toward significance. All remaining variables were not significantly associated with 25[OH]D deficiency or insufficiency. Overall, we have observed a U-shaped risk profile for both 25[OH]D deficiency/insufficiency, with higher likelihood in extremes of WC. The odds, 95% confidence intervals and significance related to particular variables are shown in Figure 4 and Figure 5. 

### 3.3. Comparison of Patients with Low Versus High Potential to UVB Exposure 

The low potential UVB exposure was found in 248 patients and high in 164, respectively. In patients with low UVBE mean serum levels of 25[OH]D were approximately 50% lower (25.2 ± 14 ng/mL vs. 37.9 ± 20.6 ng/mL *p*< 0.001). However, general functional status presented by NYHA class was better in patients with low UVBE (2.48 ± 0.83 vs. 2.64 ± 0.71, *p* = 0.04). Groups with different UVBE did not differ with respect to nutritional markers such as albumin levels (41.2 g/L vs. 41.0 g/L, *p* = 0.62), lymphocyte counts (1.85*10^3^/mm^3^ vs. 1.86*10^3^/mm^3^, *p* = 0.33), calcium (2.28 mmol/L vs. 2.30 mmol/L, *p* = 0.27) and cholesterol (4.4 mmol/L vs. 4.4 mmol/L, *p* = 0.89). Additionally, the CONUT index was similar (0.48 vs. 0.48, *p* = 0.97) in these groups. Strong prognostic markers of HF, such as NTproBNP levels, MVO_2_ and one-year death rate, did not differ between groups (2309 pg/mL vs. 2462 pg/mL, *p* = 0.25), (14.7 mL/kg*min vs. 14.5 mL/kg*min, *p* = 0.77), (14.5% vs. 12.2%, *p* = 0.50). 

## 4. Discussion

It was shown in our study that the prevalence of both 25[OH]D insufficiency and deficiency defined according to Endocrine Society [22] was high and comparable to data already published in HF [18,24]. To the best of knowledge, our study is the first where metabolic history represented by changes of dry body weight during HF, both weight gain or weight loss were correlated with the risk of low serum 25[OH]D. The novelty of this study is the demonstration of the increasing likelihood of 25[OH]D insufficiency/deficiency in subgroups with extreme weight change, particularly weight loss, during HF. The association was independent of HF severity represented by the NYHA class or other strong prognostic markers, and other known factors affecting serum 25[OH]D levels such as UVB exposure and age. These findings are in contrast to previous studies where the lower peak oxygen consumption was linked to deficient levels of 25[OH]D [25,26]. Several differences between our and these studies may provide explanation for discrepancies. Previous studies concerned patients with predominately mild HF; neither duration of HF, history of weight changes and treatment of HF were reported. Authors do not provide data on HF-induced and treatment-related metabolic response. It is widely accepted that exercise intolerance is at least in part due to catabolic predominance and consequent fat-free tissue depletion [27]. The subjective (NYHA class) and objective (MVO_2_) measures of exercise capacity should rather be regarded as downstream events initiated by catabolism. Therefore, the use of WC in prediction model of 25[OH]D deficiency/insufficiency would potentially suppress the effect of NYHA class, MVO_2_ or even NTproBNP. 

Another possible explanation would be the difference in volume control in patients included in previous studies. This important clinical issue was not addressed in previous reports, while in our patients, the completeness of decongestion was a particular focus. It was shown recently that volume overload correlated positively with serum levels of fibroblast growth factor-23 (FGF-23), known to inhibit synthesis and promote breakdown of 25[OH]D [28]. Elevated levels of FGF-23 frequently seen in patients with HF would be the link between reduced serum 25[OH]D and worse functional status and clinical outcome [29].

In agreement with previous findings [30], we also revealed a trend towards a higher risk of 25[OH]D insufficiency/insufficiency and weight gain. However, in contrast to previous studies, the risk of 25[OH]D insufficiency/deficiency was independent of body composition indices, specifically fat-free and fat mass [31]. 

Due to cross-sectional study design we cannot draw any conclusions on causality. Below, we discuss some potential mechanisms worth consideration. Generally, one of the key questions of the current vitamin D debate is whether low 25[OH]D level reflects just poor health status or if it is implicated as a mediator in a given pathology [32]. This obviously applies to research on the role of vitamin D in HF. 

Several HF-related factors may be responsible for reduced 25[OH]D serum levels. Nutritional vitamin D is minor component of body supply [33]. In HF, nutrition is usually inadequate [34,35] and anorexia occurs frequently [36]. All these factors are usually parallel in more advanced HF stages represented for example by NYHA class. Skin production of vitamin D under ultraviolet radiation is a main source of vitamin D in the body [33]. The patients with more advanced HF class have usually more muscle wasting, which is typically associated with more impaired functioning in daily life, giving rise for higher NYHA. In comparison to patients with lower NYHA class, more functionally impaired people might have likely stayed at home, thus avoiding sun exposure. These patients might have had less skin ultraviolet radiation in comparison to patients with more stable weight despite the same date of blood sampling. Taken together, the factors discussed above may well explain why patients with more weight loss may have higher probability of low 25[OH]D. 

In our cohort we attempted to find the potential source of bias related to general worse health in HF. The risk of 25[OH]D insufficiency or deficiency in higher quintiles of WC persisted even after adjustment to NYHA class, but also after replacement, this subjective measure with an objective one, such as the maximal oxygen consumption during symptom-limited treadmill exercise. Accordingly, the use of NTproBNP instead of NYHA class did not change increased risk of 25[OH]D deficiency or insufficiency. Furthermore, as nutrition quality may have a significant impact on serum 25[OH]D levels, we corrected risk with integrated nutritional index, though with no change of global risk profile. 

Our analysis showed that the strongest predictor of low serum 25[OH]D was potential UVB exposure. Patients whose serum 25[OH]D were analyzed during sunny period had approximately 50% higher levels of 25[OH]D, although they had similar markers of nutrition, similar HF risk profile and mortality, and even worse physical performance as defined by higher NYHA class. This apparent paradox is hard to explain. Differences in diets used during sunny and cloudy days that we did not take into account may be among possible explanations. 

The higher likelihood of low 25[OH]D in subgroups of either weight loss or weight gain, which is independent of UVBE, may also be explained by other factors. Apart from potential worse availability discussed above, inhibited synthesis of 25[OH]D, accelerated catabolism and increased redistribution to tissues is worth considering [37]. As we have mentioned, FGF-23 is among the known inhibitors of 25-hydroxylase and activators of 24-hydroxylase, which are a key enzymes of 25[OH]D synthesis and catabolism, respectively [28]. In HF, serum levels of FGF-23 are elevated and correlated with HF severity [29]. The reasons for FGF-23 increase in HF are unclear, but elevated serum levels of phosphate and more volume overload leading to tissue ischemia are among the most probable [29,38]. We have not measured FGF-23 in out cohort; however, patients in subgroups of more weight loss had higher serum levels of phosphorus, NTproBNP and need for more loop diuretics to secure volume balance. This may indirectly suggest elevation of FGF-23 and inhibition of synthesis and promotion of catabolism as a possible mechanism of lower levels 25[OH]D in patients with more weight loss.

In patients with weight gain, another possible mechanism of lower 25[OH]D should be mentioned. It is widely accepted that adipose tissue is the main reservoir of 25[OH]D in the body [39]. Expansion of this body compartment in patients with weight gain may be at least in part responsible for the reduction of this compound in serum. Moreover, it has been recently shown that adipose tissue expresses functional 24-hydroxylase, an enzyme catabolizing 25[OH]D into inactive 24,25-dihydroxy catabolite [40]. Hence, active degradation by expanding adipose tissue cannot be ruled out. 

Alternative explanations of lower 25[OH]D in patients with more pronounced weight loss should also be taken into consideration. Deficiency of vitamin D for any reason enables more body wasting. A number of arguments speak in favor of this concept. Vitamin D inhibits renin-angiotensin system [41], which, through AT1 angiotensin receptors, mediates vasoconstriction, oxidative stress and apoptosis [42]. All abovementioned processes are known inducers of catabolism and weight loss in HF [1]. Independently of the effects on renin-angiotensin system, activation of VDR in experimental settings was shown to inhibit inflammatory response. This includes the regulation of expression of genes for pro-inflammatory mediators, i.e., cyclooxygenases, interference with central pro-inflammatory transcription factors, for example NF-kapaB, and activation of numerous signaling pathways [43]. Of particular note is the association of low vitamin D signaling with the expression of myostatin, one of the key factors responsible for inhibition of muscle regeneration and growth [10,44]. Finally, the concept is supported by a recent meta-analysis of studies with vitamin D supplementation, which shows a reduction of serum inflammatory markers in HF by 25[OH] intervention [45]. Indeed, our patients in the top weight loss subgroup had the highest levels of hsCRP. 

During mild to moderate WC, fat-free tissue was relatively protected at a cost of more fat mass wasting. The various levels of catabolic challenge may have a different impact on either cellular composition of fat and muscle tissue or the ability of different cells for storage and breakdown of 25[OH]D. Not only muscle mass but also physical activity was shown to modulate 25[OH]D levels [46]. This potential modulator of serum 25[OH]D was not controlled in our study and was likely different depending on muscle mass and function. 

Our study has several limitations. Cross-sectional study design precluded an analysis of causality. Body weight changes after HF onset was based on medical history, not measurement, which might have been a source of significant bias. Another cause of some bias may come from the fact that samples for the measurement of 25[OH]D and NTproBNP were stored deep-frozen for various amounts of time for individual patient. The extreme difference between cases exceeded 60 months and could be responsible for the case-to-case difference due to the degraded analyte fraction before the final assay took place. The use of DEXA for measurement of fat-free mass in HF patients might also have been a source of measurement imprecision. Despite our efforts to establish edema-free status, we could not rule out water excess in some patients. If present, it may have distorted body composition analysis because DEXA technic recorded extracellular water as fat-free mass. Appetite and nutritional data were not obtained; therefore, it was not possible to estimate nutritional differences across WC subgroups. Date of blood sampling was a proxy for sun exposure. We measured 25[OH]D, which is inactive intermediate not VDR active ligand. 

### Clinical Relevance 

The active form of vitamin D has numerous anabolic, renin inhibiting and aldosterone suppressing effects; thus, the finding of 25[OH]D deficiency/insufficiency association with WC fits well into the generally-accepted concept of anabolic/catabolic imbalance and its changes following treatment over time as the main reason for body wasting in HF. If confirmed in prospective intervention studies, it may have significant clinical relevance by justifying vitamin D supplementation strategies as a useful adjunct for the prevention of catabolism related to weight loss. 

## 5. Conclusions

Metabolic instability as reflected by weight change over the time in HF, both weight loss or gain, but not nutritional and body composition indices, are independently associated with deficiency/insufficiency of 25[OH]D. The reasons for 25[OH]D deficiency/insufficiency at extremes of weight change spectrum may have different mechanisms. The identification of mechanisms underlying lowering of 25[OH]D serum levels in weight loss or gain needs further studies. 

## Figures and Tables

**Figure 1 jcm-09-01228-f001:**
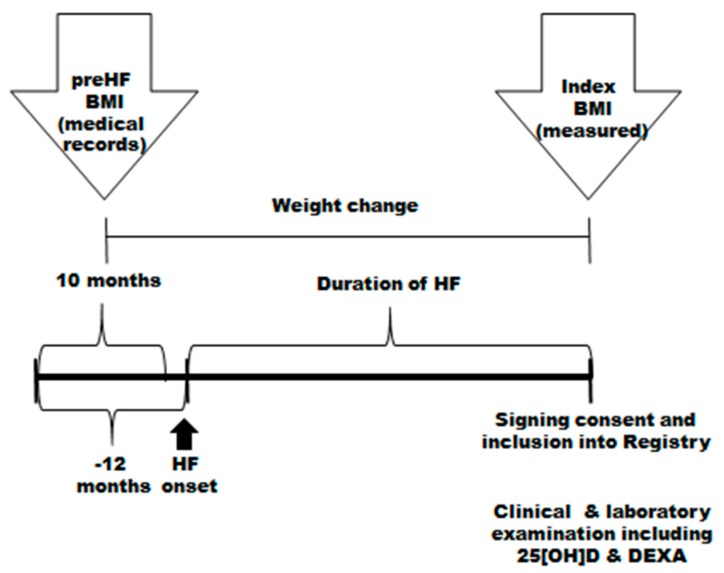
The time sequence of study procedures; BMI, body mass index; HF: heart failure.

**Figure 2 jcm-09-01228-f002:**
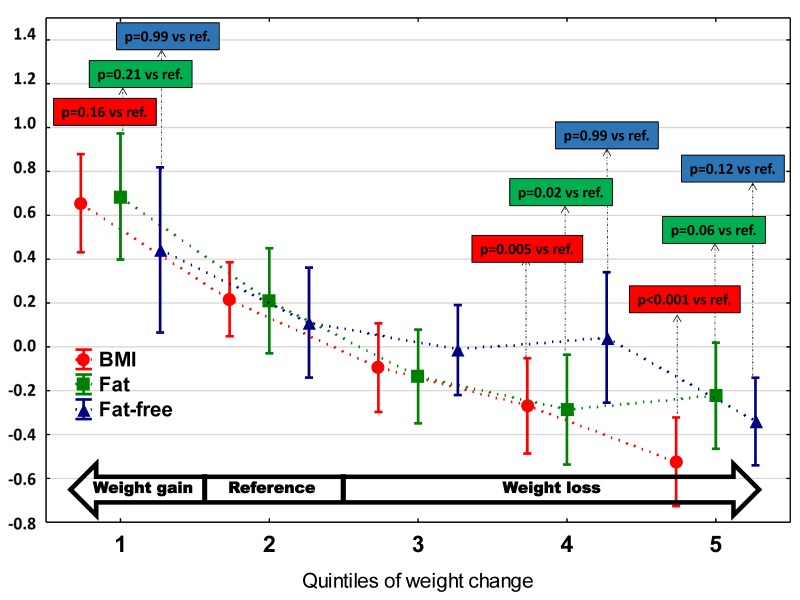
Standardized values of BMI, fat and fat-free tissue according to quintiles of WC.

**Figure 3 jcm-09-01228-f003:**
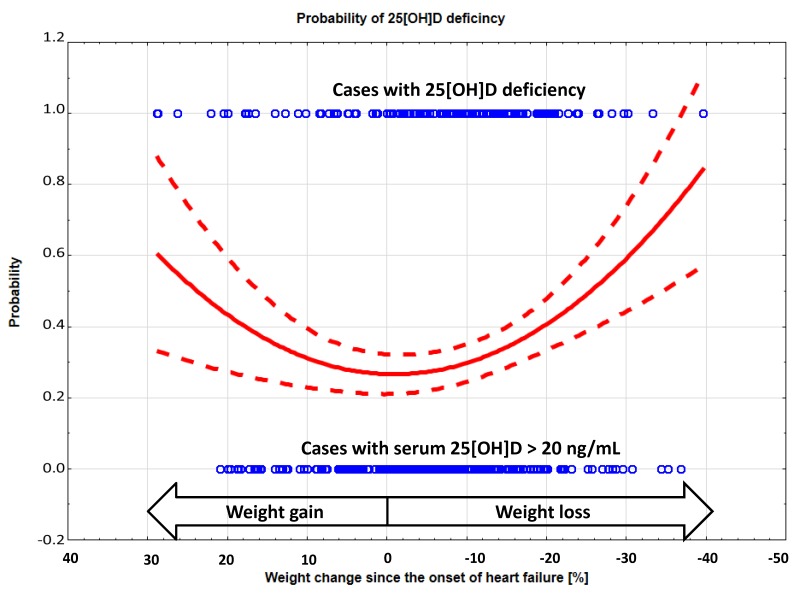
Probability and 95% confidence intervals of 25[OH]D deficiency across spectrum of WC.

**Figure 4 jcm-09-01228-f004:**
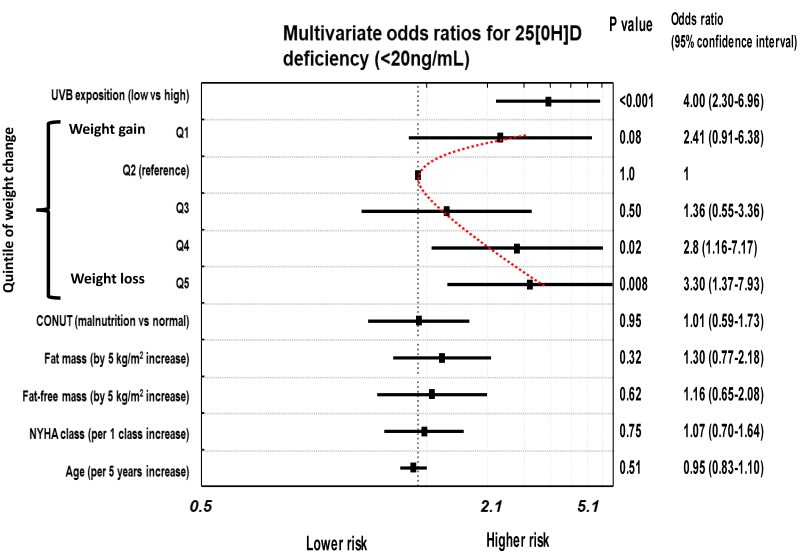
The risk of 25[OH]D deficiency in patients with different levels of weight change (WC). Independent predictors in multivariable analysis.

**Figure 5 jcm-09-01228-f005:**
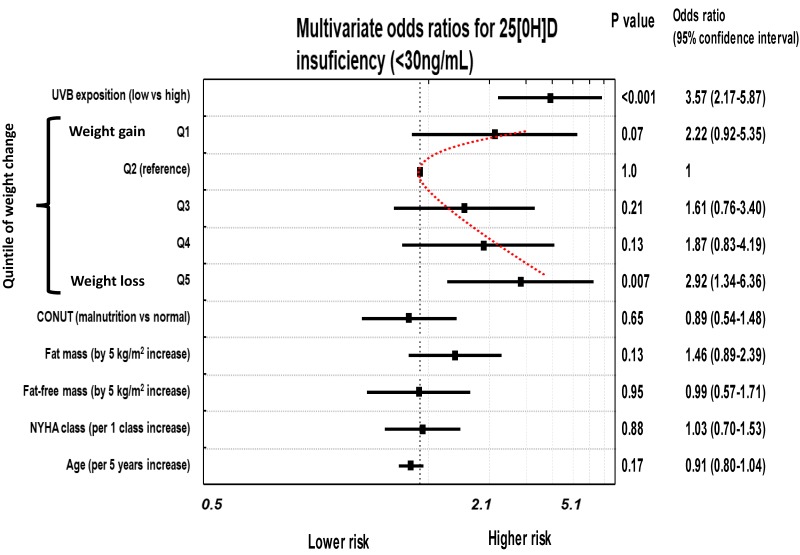
The risk of 25[OH]D insufficiency in patients with different levels of weight change (WC). Independent predictors in multivariable analysis.

**Table 1 jcm-09-01228-t001:** Clinical and laboratory characteristics of study group. Comparison between subgroups of weight changes (WC).

Feature		All N = 412	Quintiles of Weight Change					
			Weight Gain [%]	Weight Stable [%] (Reference)	Weight Loss [%]			
			Q1 N = 76 (+36.60 to +2.50)	Q2 N = 99 (+2.27 to −4.17)	Q3 N = 82 (−4.35 to −9.33)	Q4 N = 69 (−9.35 to −15.26)	Q5 N = 86 (−15.29 to −39.60)	*p*-Value
**Demography**								
Sex [% female]		14.3	17.1	5.2	7.W	11.6	19.8	0.177
Age [years]		54 (10)	54.0 (8.5)	56.0 (12.0)	54.0 (11.0)	53.0 (8.0)	55.0 (13.0)	0.79
Pre-HF BMI [kg/m^2^]		28.3 (5.8)	26.6 (5.2)	27.8 (5.9)	27.8 (5.0)	28.4 (5.5)	31.2 (7.2) and	<0.0001
Index BMI [kg/m^2^]		26.6 (5.7)	29.8 (6.0)	27.5 (5.8)	26.1 (5.0)	25.2 (5.6) @	24.5 (5.4) and	<0.0001
Weight loss in HF		−6.1 (13.3)	+8.4 (11.5) and	G−1.4 (3.0)	−6.9 (3.0) and	−12.0 (2.5) and	−19.5 (6.9) and	<0.0001
Ischemic etiology [%]		68.0	69.7	71.7	63.4	73.9	61.6	0.38
Duration of HF [months]		31 (50)	43 (17)	29 (40)	22 (56)	25 (29)	44 (69)	0.51
UVBE [low/high] [%]		60.2/39.8	63.2/36.8	57.6/42.4	59.8/40.2	60.9/39.1	60.5/39.5	0.97
**Clinical characteristics, echocardiography and body composition**								
NYHA class I/II/III/IV [%]		9/36/46/9	13/46/38/3	18/41/36/5	8/50 /38/4	1/23/62/14 and	3/19/57/21 and	<0.0001
LVEF [%]		25 (12)	27 (15)	25 (14)	24 (10)	23.0 (10)	22.0 (8.0)	<0.0001
MVO_2_ [mL/kg fat-free mass*min]		21.2 ± 7.4	22.3 ± 6.8	21.9 ± 7.3	21.0 ± 7.1	18.9 ± 6.0	21.7 ± 8.8	0.07
MVO_2_ [mL/kg*min]		14.6 ± 4.7	15.1 ± 4.2	15.1 ± 4.9	14.9 ± 4.6	13.0 ± 3.9 #	14.7 ± 5.4	0.01
Fat mass [kg/m^2^]		7.3 (3.2)	9.2 (3.3)	7.8 (2.9)	7.0 (2.8)	6.4 (2.3) #	6.7 (3.5)	<0.0001
Fat content [%]		28.7 (8.3)	32.4 (6.3)	29.8 (7.1)	28.0 (9.5)	26.2 (6.8) #	28.3 (10.6)	0.0001
Fat-free mass [kg/m^2^]		17.9 ± 2.4	19.0 ± 2.9	18.1 ± 2.4	17.8 ± 2.0	18.0 ± 2.7	17.0 ± 2.1	0.0002
**Biochemistry and nutritional indices**								
NTproBNP [pg/mL]		1376 (2645)	1034 (1226)	1092 (1778)	1370.5 (3068)	1996 (3274) and	2734 (3538) and	<0.0001
eGFR_MDRD_ [mL/min*1.73 m^2^]		88.5 (35.8)	88.4 (39.9)	87.3 (33.8)	92.8 (26.7)	89.1 (34.0)	80.9 (42.4)	0.47
eGFR_MDRD_ < 60 mL/min*1.73 m^2^ [%]		15.0	14.5	15.2	15.9	11.6	17.4	0.57
hsCRP		2.8 (5.0)	2.6 (4.0)	1.9 (3.4)	2.1 (4.0)	2.8 (5.4)	5.2. (8.8) and	0.002
Phosphate [mmol/L]		1.1 ± 0.2	1.02 ± 0.2	1.07 ± 0.2	1.12 ± 0.2	1.10 ± 0.2	1.19 ± 0.3 and	0.001
Lymphocyte [*10^3^/mm/^3^]		1.8 (0.8)	1.8 (0,8)	1.9 (0.9)	1.8 (0.8)	1.9 (0.8)	1.7 (0.8)	0.047
Albumin [g/L]		41 ± 4.0	42 ± 3.6	41 ± 3.5	41 ± 4.0	41 ± 3.8	40 ± 4.3	0.11
Cholesterol [mmol/L]		4.4 ± 1.2	4.6 ± 1.3	4.3 ± 1.1	4.4 ± 1.2	4.5 ± 1.1	4.2 ± 1.2	0.1
Calcium* [mmol/L]		2.27 (0.2)	2.23 (0.2) #	2.28 (0.2)	2.26 (0.2)	2.26 (0.2)	2.33 (0.2) @	<0.0001
CONUT [non/malnourished] [%]		51.9/48.1	59.2/40.8	55.6/44.4	48.8/51.2	58.0/42.0	39.5/60.5	0.07
25[OH]D	ng/mL	26.3 (22.6)	28.9 (22.0)	30.1 (22.5)	27.2 (18.9)	23.8 (29.3)	21.7 (19.5) #	0.01
	% < 30 ng/mL	58.5	56.8	49.5	57.3	62.3	68.6	0.11
	% < 20 ng/mL	32.5	32.9	23.2	23.2	42.0	44.2	0.004
**Comorbidities [% of patients with diagnosis]**								
Hypertension		57.7	64.5	60.6	53.7	59.4	51.2	0.42
Diabetes mellitus type 2		28.2	23.7	23.2	23.2	37.7	34.9	0.09
Hypercholesterolemia		58.0	60.5	61.6	54.9	63.8	50.0	0.34
History of smoking		74.0	72.4	71.7	78.1	68.1	79.1	0.48
**Pharmacotherapy [% of treated patients]**								
ACEI/ARB		93.7	96.1	96.0	92.7	89.9	93.0	0.48
Beta-blockers		97.3	98.7	98.0	96.3	98.6	95.4	0.61
Aldosterone antagonists		89.6	81.6	83.8	92.7	95.7	95.3	0.004
Loop diuretics		83.3	75.0	75.8	84.2	91.3	91.9	0.004
Digoxin		43.7	26.3	32.3	48.8	55.1	58.1 #	<0.0001
1-year mortality [%]		13.4	11.8	8.0	13.4	11.6	23.2	0.04

Pre-HF BMI, body mass index within a year preceding the onset of heart failure; Index BMI, body mass index at a day of inclusion into the study; NYHA class, New York Heart Association class, UVBE, ultraviolet beams exposition defined as high (if blood collected between April to September) or low (when blood was sampled between October to March); MVO_2_, maximal symptom-limited oxygen consumption on treadmill exercise; eGFR^MDRD^, estimated glomerular filtration rate based on equation from Modification of Diet in Renal Diseases Study; hsCRP—high sensitivity C-reactive protein; 25[OH]D; 25-hydroxy vitamin D; NTproBNP, N-terminal fragment of brain-type natriuretic peptide; ACEI/ARB, Angiotensin Converting Enzyme Inhibitor / Angiotensin Receptor Blockers; *, corrected for albumin concentration; #, *p* ≤ 0.05 versus reference Q2; @, *p* ≤ 0.01 versus reference Q2; and, *p* ≤ 0.001 versus reference Q2.

**Table 2 jcm-09-01228-t002:** Quintiles of WC in HF and the risk of 25[OH]D deficiency/insufficiency.

Feature	Quintiles of Weight Change				
	Weight Gain	Weight Stable (reference)	Weight Loss		
	Q1 (+36.62% to +2.5%)	Q2 (+2.27% to −4.17%)	Q3 (−4.35% to −9.33%)	Q4 (−9.35% to −15.26%)	Q5 (−15.29% to −39.56%)
	25[OH]D serum level < 20 ng/mL (deficiency), Odds ratio ± 95 Confidence Intervals, *p*-value				
Unadjusted	1.62 (0.83–3.6), *p* = 0.93	1.0	0,99 (0.40–2.00), *p* = 0.99	2.39 (1.23–4.67), *p* = 0.01	2.62 (1.39–4.92), *p* = 0.003
Model 1	2.47 (0.94–6.53), *p* = 0.06	1.0	1.38 (0.56–3.41), *p* = 0.32	2.97 (1.21–7.29), *p* = 0.02	3.42 (1.46–8.02), *p* = 0.006
Model 2	2.41 (0.91–6.38), *p* = 0.08	1.0	1.36 (0.55–3.36), *p* = 0.50	2.80 (1.16–7.17), *p* = 0.02	3.30 (1.37–7.93), *p* = 0.008
	25[OH]D serum level < 30 ng/mL (insufficiency), Odds ratio ± 95 Confidence Intervals, *p*-value				
Unadjusted	1.33 (0.73–2.43), *p* = 0.35	1.0	1.37 (0.76–2.47), *p* = 0.29	1.64 (0.78–3.45), *p* = 0.20	2.23 (1.22–4.07), *p* = 0.01
Model 1	2.53 (1.07–6.01), *p* = 0.03	1.0	1.53 (0.73–3.20), *p* = 0.26	1.71 (0.78–3.72), *p* = 0.20	2.72 (1.30–5.69), *p* = 0.008
Model 2	2.22 (0.92–5.35), *p* = 0.07	1.0	1.61 (0.76–3.40), *p* = 0.21	1.87 (0.83–4.19), *p* = 0.13	2.92 (1.34–6.36), *p* = 0.007

Model 1, adjusted for UVBE (high vs. low), CONUT (normal nutrition vs. malnutrition), fat mass, fat-free mass; Model 2, adjusted for age, NYHA, UVBE (high vs. low), (normal nutrition vs. malnutrition), fat mass, fat-free mass.

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
