# Peer review of "The Association between Serum Levels of 25[OH]D, Body Weight Changes and Body Composition Indices in Patients with Heart Failure"

_jcm, 2020, doi:10.3390/jcm9041228_

Round 1

Reviewer 1 Report

To the Authors

General Considerations

The aim of this study was to determine the association between weight/body composition changes and 25(OH)D in patients with heart failure. Authors enrolled 412 patients and fat-free mass determined by dual energy X-rays absorptiometry (DEXA) and serum levels of 25[OH]D were analysed. Logistic regression was used to calculate odds ratios for 25[OH]D insufficiency or deficiency related to changes in body weight.

The most important results of this study are that the risk of 25[OH]D deficiency / insufficiency was independently associated with potential UVB exposure, but not with nutritional status and body changes. Some unexpected results are that the indices of severity of HF (such as NYHA class and NT-proBNP) did not significantly affect the risk estimation of 25(OH)D in HF patients. These data may be in contrast with other recent studies (for example: Saponaro F. et al. Endocrine 2017;58:574-81; Porto CM, et al. ESC Heart Fail 2018;5:63-74). These unexpected results may be due to some statistical problems related to the low number of patients studied compared to the 5 sub-groups of patients and the several independent/informative variables.  Conflicting results are also reported on supplementation of vitamin D in HF patients (D’Amore C.  et al. Nutr Metab Cardiovasc Dis 2017;27:837-49; Dattilo G. et al. J AOAC Int 2018;101:939-41; Rodriguez AJ. et al. Sci Rep. 2018 Jan 18;8(1):1169). Authors should discuss these important issues in more details.

I have also to address to the Authors some specific points in order to try to improve the scientific message of this study.

Specific Points

  1. Laboratory tests. Authors state that patients were recruited in outpatient clinic between January 2004 and March 2013. Authors reported that commercially available reagents (Roche Diagnostics, Switzerland) were used to measure several biomarkers, also including NT-proBNP. Authors should clearly state when these laboratory tests were performed. It is important to know whether the NT-proBNP assay was performed as soon as after enrolment of patients or, on the contrary, serum/plasma samples were collected and then stored for many months or years before the assay. Indeed, Authors state that blood was immediately centrifuged at 4°C and stored at -75°C for further analyses, but the time of storage is not clearly reported. Natriuretic peptides, including NT-proBNP, can be degraded in vitro even at 75° C (Clerico A. et al. Clin Chim Acta 2015;443:17-24). Another important issue is that in 2009 Roche Diagnostics changed the ECLIA NT-proBNP assay by introducing a more performant immunoassay using two monoclonal antibodies instead of the old method with polyclonal antibodies (see Prontera C. et al. Clin Chim Acta 2009; 400:70-73). Authors should clearly report when the NT-proBNP was performed and the type of ECLIA NT-proBNP assay was actually used and when the assay was performed. If two different NT-proBNP methods were used throughout the study, this information should be added in the Material and Methods section or in the Discussion section of limitations of the study with the reference reporting the different quality specification of the two methods (i.e,, Prontera C. et al. Clin Chin Acta 2009; 400:70-73). Authors should better describe the NT-proBNP assay and also report an estimation of the storage time in order to evaluate the possible in vitro degradation of samples.
  2. Cardiac cachexia (see references 1,2). Authors should briefly discuss and also report more information on the pathophysiological problems related to the so-called paradox of low and also sex-related BNP levels in obesity and in patients with HF (for more recent articles on this important point: Clerico A. et al. Heart Fail Rev 2012;17:81-96; Clerico A. et al. Eur J Heart Fail 2018; 20:1215-16).

Author Response

  1. We thank You for your important concern regarding data of laboratory tests and type of tests used. In fact, data shown in the manuscript refers to Prospective Registry of Heart Failure (PRHF) from which we have derived our cohort. For current analysis patients were recruited from August 2004 to July 2010. All standard laboratory tests such as creatinine, CRP, cholesterol etc. were carried out at a day of inclusion to PRHF. Additional analyses dedicated to current analysis such as NTproBNP and 25[OH]D, were performed later (in September 2011) on frozen samples.
In obedience to your suggestions we included in the manuscript a detailed information about exact dates of particular measurements. We showed maximal, minimal and median storage time and also provided characteristics of laboratory tests used to measure NTproBNP and 25[OH]D.
As the time of storage varied giving rise for potential bias, we included a proper explanation in limitation section of the manuscript.    

  1. In our analyses of 25[OH]D predictors, apart from weight changes, we also needed heart failure severity markers. The most important one is mortality. In our case, 1-year-mortality was the lowest in weight stable patients (Q2) and increased both in patients losing and gaining weight .       
Obesity paradox refers to survival advantage of higher BMI in many chronic conditions including heart failure. In obese patients with heart failure, natriuretic peptides are lower in comparison to normal weight with similar clinical stage of heart failure. In fact, levels of natriuretic peptides are more often associated with fat-free mass, than with fat percent or BMI in healthy subjects (Sandeep R. Das at al.: “Impact of Body Mass and Body Composition on Circulating Levels of Natriuretic Peptides Results From the Dallas Heart Study”. Circulation. 2005;112:2163-2168), and in acute illness (Fang-Yang Huang at all.: „The influence of body composition on the N-terminal pro-B-type natriuretic peptide level and its prognostic performance in patients with acute coronary syndrome: a cohort study”. Cardiovasc Diabetol (2016) 15:58).
The phenomenon of obesity paradox complicates diagnosis of heart failure, but  serum levels of natriuretic peptides still retain its prognostic power (Lutz Frankenstein at all.: “Relation of N-terminal pro-brain natriuretic peptide levels and their prognostic power in chronic stable heart failure to obesity status”. European Heart Journal 2008; 29, 2634–2640).
The modulatory effect of sex and some aspects of body composition was shown in population without heart failure and NTproBNP levels generally < 100 pg/mL (Navin Suthahar at all: “Sex‐specific associations of obesity and N‐terminal pro‐B‐type natriuretic peptide levels in the general population” European Journal of Heart Failure 2018;20:1205–1214). These data does not necessarily apply to patients with NTproBNP levels at a range of 1000 – 2000 pg/mL as in our study.
Oxygen consumption at peak exercise belongs to objective measures of exercise capacity and it is known to be closely linked to heart failure severity and prognosis. To show  the effect of various fat contribution in weight change subgroups we calculated and added in Table 1 MVO2 values expressed in kg of fat-free tissue. In multivariable analysis body composition indices were included, so variable of fat content was taken into account.
NYHA class is a subjective measure of exercise tolerance but it is widely accepted as severity and prognostic marker in heart failure.
In multivariable analysis we replaced NYHA for MVO2 and then for NTproBNP with similar effect, that is no independent association with either deficiency or insufficiency of 25[OH]D.
We agree that in literature there are reports with conflicting results. In Discussion we added a paragraph where we tried to point out potential reasons explaining discrepancy. In Table 1 we also added data regarding CRP levels in weight change subgroups and discussed them in the light of results of vitamin D supplementation.
Finally, we agree that low number of patients in weight change subgroups, especially in Q4 may have an impact on results of multivariable analysis. This also has been included in limitation section.

One more time we would like to thank both Reviewers for their help to improve the scientific quality of our manuscript.

Sincerely yours
Apolonia Stefaniak MS,
Piotr Rozentryt MD, PhD

Reviewer 2 Report

Introduction:

Line 72: add vitamin D levels. 

Line 74-74: add a sentence like: Therefore, the aim of this study was ....

Methods:

The recruitment of the patient is very widely so this can have an effect for example in the use of the different drugs. In 2004, drugs were different from 2010. How do you work with that? It could be added as a limitation.

Measurements:

Add reference for indexBMI, WC and LVEF formulas

Do you have an echo to verify CHF?  

Why do you group according UVB, do you have a reference? Do you try by 4 seasons?

If you have information about Kidney disease (it is a stronger predictor in this population), you should add in characteristics and adjusted models. It also influences the homeostasis of Calcium and Vitamin D

Any information about: pacemaker or coronary artery bypass?

Alcohol intake? 

In table 1, add vitamin D levels as continuous measure.

In the adjusted models, duration of the CHF/HF, Kidney disease, CAD, creatinine and GFR should be tested. 

Author Response

  1. We would like to thank You for your suggestions. All were crucial and important that’s why we have revised our manuscript according to them.
We have corrected a mistake in line 72 in the manuscript and added a sentence you proposed in Aim section.
  1. Treatment recommendations, at least for pharmacotherapy, did not change between 2004 and 2011. Ivabradine has been recommended since 2012. As shown in Table 1 nearly all patients were treated with ACEI/ARB, beta-blockers, aldosterone antagonists and diuretics. Differences in treatment with aldosterone antagonists, digoxin and diuretics appear to be due to clinical situation rather than to guidelines. So, we do not feel that mentioning this in limitation section is really necessary.
  2. All references for indexBMI, WC and LVEF formulas were added in the manuscript. .
  3. Echo was performed in all the patients and detailed information is provided in lines 124-127.
  4. We have decided to group the patients according to UVB potential exposition following research showing insufficient production of vitamin D3 between October to March in geographical altitude of Poland. We put a proper reference in bibliography (ref. 23 in corrected version). According to these data we did not attempt to make a division by four seasons.
  5. In fact, kidney disease is strong factor modifying metabolism of vitamin D. In our cohort, median of eGFR in all subgroups was >80 ml/min which we think does not substantially impact metabolism of vitamin D. However, following your suggestion, we included additional line in table 1 where we show percentage of patients within each subgroup having eGFR < 60 ml. We have also tested kidney function as a univariate predictor of either deficient or insufficient 25[OH]D levels showing lack of correlation (not shown in the manuscript).
  6. All patients with coronary artery disease as a dominant reason were checked before inclusion for the need for revascularization. In case of revascularisation they underwent it but we do not have detailed information on revascularization method. Data on pacemaker and alcohol intake were not available.
  7. In Table 1 data on 25[OH]D were provided as continuous variable and as percentage deficient / insufficient.
  8. As in case of kidney function, duration of heart failure and its aetiology was tested in univariate model with p-value > 0.2 and were not included into multivariable model in order to avoid unjustified overfitting.

One more time we would like to thank both Reviewers for their help to improve the scientific quality of our manuscript.

Sincerely yours
Apolonia Stefaniak MS,
Piotr Rozentryt MD, PhD

Round 2

Reviewer 1 Report

Authors revised the manuscript according to the suggestions made by the Reviewer. The manuscript is now improved.